# Probing sustained attention and fatigue across the lifespan

**Simon Hanzal** [1]*, **Gemma Learmonth** [1,2], **Gregor Thut** [1], **Monika Harvey** [1]

**1** School of Psychology and Neuroscience University of Glasgow, Glasgow, United Kingdom, **2** Faculty of Natural Sciences, University of Stirling, Stirling, United Kingdom

\* hanzalsimon@gmail.com

## Abstract

Trait fatigues reflects tiredness that persists throughout a prolonged period, whereas state fatigue is a short-term reaction to intense or prolonged effort. We investigated the impact of sustained attention (using the SART) on both trait and state fatigue levels in the general population. An online version of the SART was undertaken by 115 participants, stratified across the whole adult lifespan. While pre-task trait fatigue was a strong indicator of the initial state fatigue levels, undergoing the task itself induced an increase in reported subjective state fatigue, and an accompanying reduction in subjective energy rating. Consistent with this finding, greater subjective state fatigue levels were associated with reduced accuracy. In addition, age was the best predictor of inter-participant accuracy (the older the participants, the greater the accuracy), and learning (i.e., task duration reducing reaction times). Moreover, a ceiling effect occurred where participants with higher trait fatigue did not experience greater state fatigue changes relative to those with low trait scores. In summary, we found improved accuracy in older adults, as well as a tight coupling between state fatigue and SART performance decline (in an online environment). The findings warrant further investigation into fatigue as a dynamic, task-dependent state and into SART performance as an objective measure and inducer of fatigue.

## Introduction

Fatigue is one of the most common symptoms experienced by people with a range of clinical conditions, for example, post-stroke fatigue affects up to 50% of stroke survivors [1]. Fatigue has also been estimated to affect up to 17% of the general population [2]. In the clinical populations, definitions of fatigue vary and are often specific for the respective populations in which they occur. Post-stroke fatigue for instance has been described as a subjective lack of physical or mental energy (or both) that is perceived by the individual to interfere with usual or desired activities with the closely related "chronic fatigue" described as a negative whole-body sensation, not proportional to recent activity [3]. There is similar variability in defining fatigue within the general population. Researchers either extend the definition from a particular syndrome, typically chronic fatigue syndrome [4] or frame fatigue more generally within experimental cognitive research, e.g., as a lapse in sustained attention [5]. One frequently adapted

**Data Availability Statement:** All experimental data files and the analysis and data collection scripts are available from the osf.io database (DOI 10.17605/OSF.IO/YNM9H).

**Funding:** SH was supported by the Economic and Social Research Council Grant: ES/P000681/1.

https://www.ukri.org/councils/esrc/ GM was funded by the Wellcome grant 209209/Z/17/Z. https://wellcome.org/ The funders had no role in study design, data collection and analysis, decision to publish, or preparation of the manuscript.

**Competing interests:** The authors have declared that no competing interests exist.

model describes fatigue as a change from baseline state in response to either physically or mentally challenging tasks, which induces a depletion of cognitive resources and lowered vigilance [6]. Another mechanism of fatigue induction could be via boredom, where repetitive and unstimulating tasks lead to an inability to maintain sustained attention [7].

Trait fatigue is measured both in clinical studies [3, 8, 9] and the working population [10–12]. It has been characterised as an innate tendency to exhibit fatigue [13]. Long-term trait fatigue depletes the ability to readily engage in moderately demanding tasks [14]. Alternatively, measures used for self-reported assessment of trait fatigue are comprised of recalled experiences of fatigue over specified time windows and dimensions (e.g., the multidimensional fatigue inventory, MFI [15]). This type of recent, self-reported fatigue has also been described as "prolonged state fatigue" [13] in order to distinguish it from an innate tendency to become fatigued (sometimes also described as "trait fatigue"). Short-term state fatigue, on the other hand, is a more transient mental state [16]. It undergoes dynamic shifts throughout the day, based on external factors corresponding to undertaken activities and tasks [17]. States are prone to shift, and this is reflected in performance changes across tasks that require sustained focus or attention [18]. State fatigue can be studied subjectively through self-report measures that are designed to capture subjective experience at any given time. Many researchers use a simple, one-item measure of subjective state fatigue [19] to link reported momentary fatigue to the objective task performance [20–22]. However, measures of state fatigue comprising several items would offer greater construct validity [23]. There has been a tight coupling of state fatigue with energy, where several studies [13, 24, 25] utilised combined measures of state fatigue comprised of energy and fatigue subscales. Although the two scales seem to be closely related, findings of divergent changes in both suggest that they constitute two related, unipolar aspects [13, 26]. In relation to fatigue, energy was described as the potential to carry out mental or physical activity [26]. The two measures can often share a similar pattern of response to fatiguing tasks [25], but have also been found to differ, especially in the context of physical activity [26, 27]. Accordingly, a more in-depth investigation of state fatigue benefits from inclusion of both of the two separate subscales. Furthermore, few studies have investigated changes of subjective state fatigue during effortful tasks [28], and it is also unclear how fluctuations in subjective state relate to objective changes in task performance. Tests of state fatigue with attentional paradigms suggest that the ability to concentrate for a prolonged period decreases over time [29, 30]. Therefore, a fatiguing task is perhaps the most immediate exogenous influence on state fatigue over and above the initial baseline stemming from trait fatigue measures. Furthermore, coupling the changes in (subjective) state fatigue with task performance would enable a direct link between (objective) reduced task performance and (subjective) fatigue measures.

## SART and fatigue

It is known already that tasks that require continuous and maintained mental effort are likely to elicit changes in fatigue [18]. A self-directed maintenance of cognitive focus [31] can be characterised as sustained attention, and a frequently researched task that relies on sustained attention is the sustained attention to response task (SART) [32–34]. The task is characterised by frequent go trials where a response is expected and rare no-go trials which tend to elicit errors arising from attentional lapses. Performance on the SART is thus typically measured in terms of response accuracy with a focus on commission errors (i.e., when erroneous responses are made in no-go trials), reaction times, and standard deviations in reaction times. The sensitivity of the task to fatigue lies in its tendency to provoke unintended motor response commission errors, with lapses in attention. Regarding trait fatigue, an initial comparison of the SART

with the cognitive failures' questionnaire [35] showed a modest negative relationship [32]. The questionnaire was principally developed to reflect trait predisposition to attentional lapses, yet investigations into larger and more diverse populations with alternative procedures and methods of analyses have shown limited support for this association [30]. However, measures relating to state fatigue have been easier to link to task performance, and indicated some change over time [36]. Thus, the SART may be a reliable means of measuring as well as experimentally inducing changes in state fatigue levels [37] while detecting whether these are related to trait fatigue.

## Age and the SART

At present there is a gap in our current understanding of fatigue across the healthy adult life span. Somewhat counterintuitively, surveys recurrently suggest fatigue to reduce with advancing age [2, 4, 12, 38]. Yet, aging has also been noted to lead to deficits in attention [39], difficulty in attentional switching [40] and lowered task-related attentional improvement [41]. On the other hand, both McLaughlin et al. [42] and Staub et al. [43] reported higher SART retention of accuracy with more advanced age, and a task closely resembling the SART showed stability of commission error rates across different age groups [44]. This goes in opposition to older adults reporting deficits in sustained attention [45] while keeping lower mind-wandering levels [34, 46] and occasional evidence for a gradual decline both in reaction times and accuracy in subsets of the ageing population in a version of the SART [47]. Differences in type of fatigue assessed may reconcile these diverging findings, and we thus investigated SART performance and trait and state fatigue systematically across the life span.

## Aims

Recently, interactive behavioural experiments have been moved online to platforms such as Qualtrics [48–51]. While these studies were of particular interest due to the increased risks of conducting face-to-face laboratory experimentation during the global pandemic, the online implementation of cognitive experiments has been shown to achieve precision comparable to the laboratory environment, whilst providing researchers access to wider, more diverse demographic groups [52]. We leveraged the online approach in the present investigation and hypothesised that in an online experiment, trait fatigue could either negatively affect task performance directly, or that trait fatigue would predispose participants to higher levels of pretask state fatigue. In turn, we proposed that this pre-task state fatigue would cause further intask changes to state fatigue, and consequently to performance on the SART. We investigated this by first recording trait fatigue measures, using a subjective self-report questionnaire. Participants then provided their momentary, pre-SART state fatigue through a subjective selfreport measure, performed an extended version of the SART and then reported their postSART state fatigue. In relation to our pre-registered protocol (available at https://osf.io/hzwvp), we specifically aimed to:

1. Investigate the correlational relationship between changes in state fatigue and performance changes on SART over time. Specifically, that no-go accuracy will decrease, and reaction times will increase as a) state fatigue increases, and b) state energy decreases.

2. Assess the relationship between no-go trial accuracy and reaction time on the SART and reported trait and state fatigue. Specifically, we predict that no-go accuracy will be lower, and reaction times will be slower in participants with a) high trait fatigue, b) high state fatigue and c) low state energy.

3. Determine the relationship between subjective trait fatigue and state fatigue, as well as changes in state fatigue as a result of the task. Specifically, that trait fatigue would be a) positively correlated with pre-task state fatigue and b) negatively correlated with pre-task state energy. We also expected to observe greater SART-induced c) increase in state fatigue when pre-task trait fatigue was high, and d) decrease in state energy when pre-task trait fatigue was high.

4. Carry out the research project online, targeting the general population across the whole lifespan and so test the viability of an online environment for general research on sustained attention. Based on previous divergent findings, we expected to observe difference in a) no-go accuracy and b) reaction times with increasing age.

## Materials and methods

### Participants

We ran a power analysis on the largest anticipated test to be performed, a two-sample independent t-test with an expected power $(1-\beta)$ of 0.80, $\alpha = 0.05$ and an expected Cohen's d of 0.4 [53]. As there was no prior evidence as to how state fatigue may relate to the other proposed measures, we powered the sample size calculation for a medium effect size based on a pilot study with 10 participants (see methods section). The power analysis was conducted using the "pwr" in R [54], determining that a minimum of 100 participants was required. Based on the pilot study (see below), we expected a drop-out rate of 10%, and because we wanted to recruit 6 stratified age cohorts of equal size, the total number of participants recruited for the study was raised to 120.

Participants were recruited from the general adult population (age $\geq$ 18) through the online platform Prolific [55]. They were recruited in 6 equally large, stratified age cohorts (18–29, 30–39, 40–49, 50–59, 60–69, 70+) to compensate for overrepresentation of younger participants in the Prolific participant pool. All participant data was acquired within 24 hours of the study portal opening on 28[th] of March 2021 at 8am (BST). Most participants carried out the task within the first two hours of its publication on Prolific (84%). Participants were admitted to the experiment if they indicated at least moderate proficiency in the English language and reported an absence of any cognitive or neurological conditions or uncorrected vision.

The study was approved by the University of Glasgow College of Science and Engineering Ethics committee (Approval number: 300200069). Digital consent was obtained from all participants by completing an online checkbox form.

### Measures

The Visual Analogue Scale for Fatigue (VAS-F [56]) was used as the state fatigue measure (see appendix). It captures changes in subjective state fatigue through 18 items divided into two subscales: one for fatigue (13 items) and one for energy (5 items), with scores from 0 (no fatigue) to 100 (maximum fatigue). It has shown excellent reliability of $\alpha = 0.93$ and $\alpha = 0.91$ for the two scales, respectively [56]. Two items of outdated language were replaced by closest possible synonyms: "worn out" to "drained", and "bushed" to "run down" to avoid repetitiveness and a poor understanding of the items. Subsequent internal consistency tests confirmed no impact on the measure's reliability.

The MFI was used to acquire trait fatigue measures [15]. This scale has been used to measure fatigue in a variety of settings and age groups and is comprised of 5 subscales with 4 items each (20 items in total) on a 5-point Likert scale. It is the most comprehensive measure to date

combining many aspects of trait fatigue in one larger scale. It has a reliability of $\alpha = .84$. and shown to lack floor and ceiling effects as well as item redundancy [57].

## Task

The SART is a task in which participants react to numerical stimuli presented in rapid succession in the middle of the screen. The general population has response times ranging from 300 to 400ms [58, 59]. We expected our data to contain time offsets of about 30ms due to the online nature of the task, expected from hardware (keyboard sampling, keyboard cable) and software (operation system, web-browser) differences (compared to a stable laboratory environment [51]). In the conventional SART, the typical standard deviation is reaction times between 50ms to 100ms. We expected a greater standard deviation in response times in our online sample in comparison to the conventional experiments, due to possible lower accuracy and variability of the devices the participants could use to access the task [51]. Whilst a very high accuracy rate was expected on the go trials ($> 90\%$), we expected no-go accuracy rate to vary greatly across participants [3, 58].

A custom implementation of the SART using JavaScript code, relying on the jsPsych package [60] hosted on an external, secure server was used to run the experiment. There was a practice block of 36 trials, followed by four blocks of 117 trials, each with a break (timed by the participants) between each block, 502 trials in total. This number was chosen to achieve a duration of around 10 minutes for the experimental part and around 20 minutes for the whole study (based on the prior piloting). Each trial consisted of a number between 1–9 presented in the centre of the screen at a 64-pixel size for 250ms. The number disappeared for 900ms before the next one was presented. Participants were instructed to respond to any number apart from the number 3 by pressing the space bar (go), whilst withholding their response for the number 3 (no-go). They were asked to balance speed and accuracy in their responses as both were used as measures of performance. For each participant, the numbers were sampled randomly, with each number appearing the same number of times, and all numbers were distributed evenly. Altogether, the no-go stimuli appeared 56 times in total, representing 11.11% of the presented stimuli.

## Procedure

Upon receiving the notification of a new study available on Prolific, participants were redirected to a survey web portal on Qualtrics [48]. The platform was chosen because it follows strict ethical protocols for participation and provides access to a stratified participant demographic [61]. Participants were first introduced to the experiment and asked for their consent. They then provided their basic demographic information, self-reported information about possible visual deficiencies and other conditions that would impact their performance in the experiment, they then filled in the MFI and VAS-F. The Qualtrics platform then performed call-backs to a server hosting the JavaScript code, forming a pop-up within the Qualtrics survey in which participants carried out the experimental tasks. They underwent a practice session for the SART, completed four blocks of the SART with breaks between them and then provided their VAS-F again. Finally, they were debriefed. To avoid expectation bias in the practice block, participants were only given general feedback about their accuracy without specified desirable outcomes.

## Statistical analyses

All data analysis was carried out in R [62] using the packages 'tidyverse' [63], 'psych' [64] 'moments' [65] and 'lm.beta' [66]. Further packages used for graphical depiction were: 'ggpubr' [67], 'viridis' [68] and 'Cairo' [69].

The behavioural data was pre-processed to acquire accuracy scores both for go and no-go trials. Go trial responses classed as anticipation errors (< 150ms) [59] were discarded. Participant reaction times were log-transformed to normalise the distribution of the residuals of the subsequent models. As per the pre-registered protocol, participants with responses that fell into either of two anticipated deliberately erroneous approaches to completing the experiment were removed from further analysis: One was responding to all trials and withholding responses randomly at the rate of the occurrence of the no-go stimuli (> 89% go stimuli correct and < 11% no-go stimuli correct), the other to randomly respond at the rate of occurrence of the no-go stimuli (< 11% go stimuli correct and >89% no-go stimuli correct), in any of the four experimental blocks. Participants who did not complete all the blocks were likewise removed from the analysis. Finally, participants showing more than one failure to correctly answer the attention check questions were excluded from the analysis also.

Correlation matrices were acquired for the five subscales of the MFI and then used to compute the Cronbach's alpha [70]. Cronbach's alpha scores were also obtained for the pre- and post-test levels of the VAS-F.

State fatigue change was obtained by subtracting the pre-task VAS-F score on both the fatigue and energy subscales from the post-task VAS-F score on both of those scales. Accuracy change scores were acquired by subtracting the no-go accuracy score in the last block from the no-go accuracy in the first block.

### Pilot study

A pilot study with 10 participants was conducted to determine the viability of the online environment for conducting a SART experiment. These results also informed the power calculation and resulted in a formulation of the accuracy thresholds for the SART. In response to the pilot, we also implemented attention checks to ensure that participants fully attended to the questions. These were in the form of an extra item on the MFI, pre-task VAS-F and post-task VAS-F asking the participant to answer with a specific numeric value. In line with the platform recommendations, participants were excluded from data analysis and refused payment if they failed more than one of the three checks.

## Results

### Exclusions

Four participants failed more than one attention check and so were not included in the sample. A further 16 participants (11.4%) attempted the task but stopped without finishing. Further participants were recruited in their place until the complete sample size of 120 was achieved. For the final analysis a total of 5 participants were excluded from the sample: Two participants experienced an unknown technical fault, two were removed for failing to achieve the minimum SART performance and one was removed for reporting a lack of sufficient English language knowledge. This left the total of participants at 115 (95.83% of complete total recruited), see Fig 1.

### Participants

A total of 115 participants were analysed after exclusions. The sample was comprised of 61 women and 54 men (46.96%), with a mean age of 48.43 (SD = 18.08) equally represented across the stratified adult age lifespan (range = 18–81). The sample was comprised of 27 different nationalities, predominantly European (n = 100, 86.96%), with the most frequent being British (n = 51, 44.35%), Polish (n = 12, 10.43%) and Italian (n = 7, 6.09%). This also meant

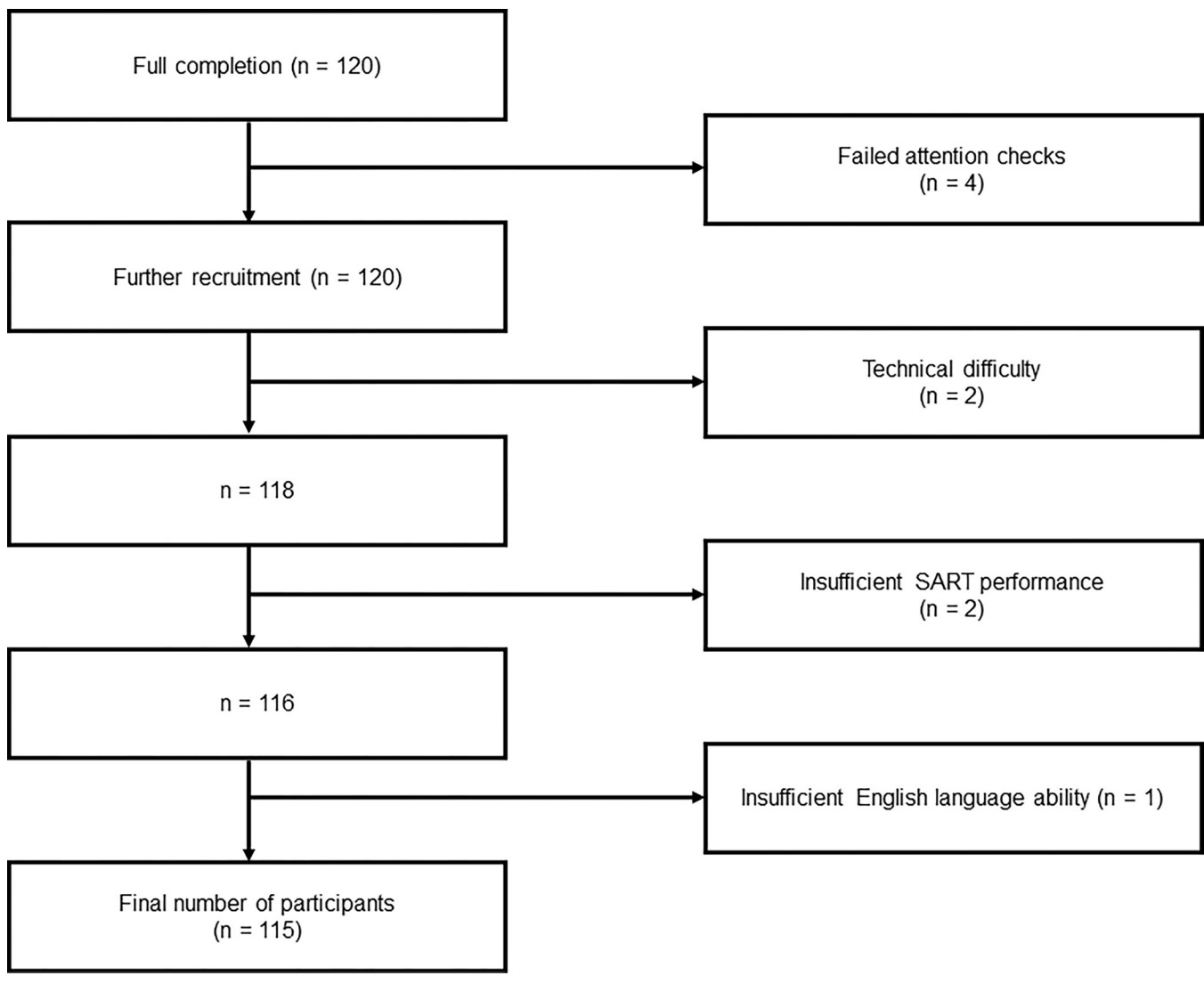

**Fig 1. Flowchart depicting the exclusion process in the study.**

that since most of the participants who completed the study accessed it from European countries under either British Summer Time (BST) or Central European Summer Time (CEST), the predominant majority of participants completed the study in the morning. The sample was socio-economically diverse, including 26 (22.61%) individuals with no higher education, 26 (22.61%) individuals with college-level or vocational training, 35 (30.43%) individuals with an undergraduate degree and 28 (24.35%) individuals with postgraduate and/or higher degrees. Sixty (52.17%) individuals were in some form of employment, whilst 13 (11.30%) comprised students and 42 (36.52%) were unemployed, retired, on leave or furloughed. No participants reported complications related to eye-conditions or cognitive difficulties which could have impeded performance. Finally, 32 (27.83%) participants reported that they currently had, or suspected that they had had, Covid-19 infection. However, only 5 (4.35%) reported a definite diagnosis with Covid within the previous 3 months.

## Questionnaires

The participants gave both their pre-task fatigue (M = 364.49, SD = 284.82) and energy (M = 264.06, SD = 114.01) scores as well as their post-task fatigue (M = 415.27, SD = 325.04) and energy (M = 241.08, SD = 118.87) scores. Change scores were thus obtained by subtracting the pre-task score from the post-task score for fatigue (M = 50.78, SD = 186.14) and energy (M = -22.98, SD = 91.76). As the change scores included post-task state fatigue levels in their calculation, we decided not to consider post-task scores separately, contrary to what we indicated in the pre-registration.

Pre-test fatigue scores predicted post-test fatigue scores, $F_{(1, 113)}$ = 234.70, p < .001, $R^2$ = .680. Likewise, pre-test energy scores were predictors of post-test energy scores, $F_{(1, 113)}$ = 102.80, p < 0.001, $R^2$ = .480, showing consistency within participants across the two time points.

The participants completed the MFI as a measure of trait fatigue. Subscale and total scores were calculated. The sample had a mean score around the midpoint of the scale (M = 51.43, SD = 13.82, range = 20–87). Even on the subscale level, the MFI showed a similar means for general fatigue (M = 10.77, SD = 3.43, 4–19), physical fatigue (M = 10.54, SD = 3.56, 4–20), mental fatigue (M = 10.12, SD = 3.59, 4–20), reduced activity (M = 10.28, SD = 3.36, 4–20) and reduced motivation (M = 9.71, SD = 3.15, 4–18). The results were similar to those found in other populational studies [17, 29] as opposed to results with means above the threshold score of 60 [71] indicative of a clinically fatigued population.

Cronbach's alphas of the MFI, as well as of the pre-test VAS-F and the post-test VAS-F, were obtained. We expected a Cronbach's alpha of 0.8 on all the scales [72]. We found a Cronbach alpha of 0.92 for the MFI total, 0.78 for general fatigue, 0.81 for physical fatigue, 0.81 for reduced activity, 0.70 for reduced motivation, and 0.86 for mental fatigue. The pre-task state fatigue scale (VAS-F) showed an alpha of 0.95, energy (VAS-E) was 0.95. Overall pre-task was 0.96. Post-task overall was 0.97, fatigue 0.97 and energy 0.95. Except for general fatigue and reduced motivation, all values reached 0.8, implying that the items in the questionnaires were internally consistent. The VAS-F showed very high internal consistency.

## SART

After removal of the trials with very short reaction times (< 150ms, 3.0% of data), and although the sample mean reaction times matched the times expected based on prior studies (364.47ms, SD = 5.15ms), they appeared not to be normally distributed (they had a heavy rightward skewness (2.02) and were leptokurtic (12.44)). So, the reaction times were corrected by log-transformation, leading to more acceptable skewness (0.50) and kurtosis (4.54). The mean within-participant standard deviations in reaction times were 99.87ms (SD = 7.76) and normally distributed (skewness = 0.22, kurtosis = 2.69). No-go accuracy was 68.23% (SD = 15.84%) and go accuracy was at ceiling, 98.86% (SD = 3.07%).

Accuracy change scores (M = -3.34%, SD = 18.82%) were acquired by subtracting the nogo accuracy score in the last block of the task, from the first block. This showed no skewness (-0.19) and no kurtosis (2.96). A multiple linear regression was run to predict no-go accuracy change from each of the five individual subscales of trait fatigue, change in reaction time, change in state energy, change in state fatigue and interaction between change in state fatigue and change in state energy. The overall model was significant $F_{(10, 104)}$ = 2.51 p = .010, $R^2$ = .190 and only fatigue change was found to be a predictor of accuracy change with a negative relationship between accuracy change and fatigue change, β = -.359, t = -2.82, p = .006, showing a small to medium effect size [53]: the greater the fatigue change, the larger the drop in accuracy across the blocks. While fatigue change was associated with accuracy change, energy

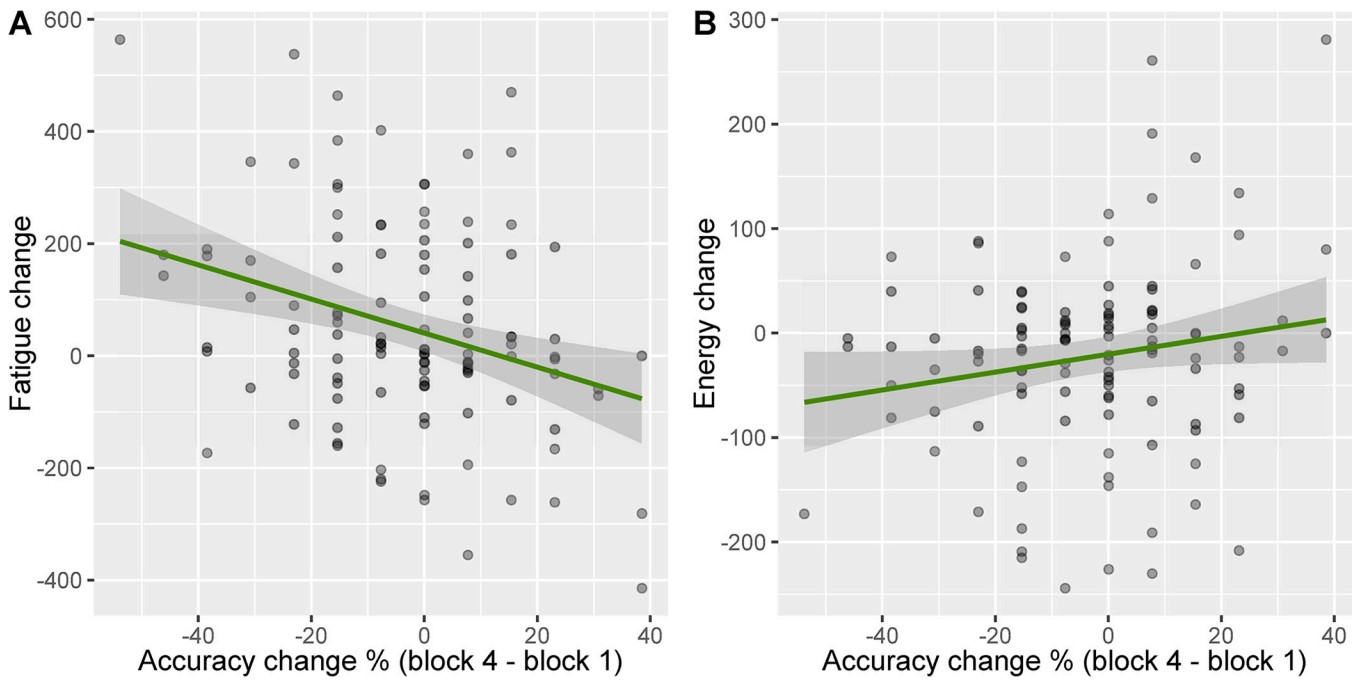

**Fig 2. Linear relationship between accuracy change, fatigue change and energy change.** (A) Relationship of accuracy change between the last and the first block and fatigue change before and after the task with 95% confidence intervals and (B) accuracy change between the last and the first block and energy change before and after the task with 95% confidence intervals.

change and the other variables were not significant. Fig 2 depicts both relationships to fatigue and energy. The variance inflation factor remained under 3 for all model variables. The full regression table is available in the Supporting Information (Table in S1 Appendix).

A multiple linear regression was run to predict no-go accuracy from block number, age, total MFI score, pre-task state energy score, pre-task fatigue score and interaction of all the subjective fatigue measures. The overall model was significant F(8, 451) = 9.40 p < .001, R2 = .140. Only age was found to be a predictor of accuracy, β = .372, t = 3.64, p < .001: the older the participants were, the more accurate they were at withholding no-go responses during the SART. The relationship between age and accuracy is depicted in Fig 3. The variance inflation factor remained under 3 for all model variables.

A multiple linear regression was run to predict correct go-trial reaction time from block number, age, total MFI score, pre-task state energy score, pre-task fatigue score and the interaction of all the subjective fatigue measures, yielding an overall significant model, F(8, 451) = 11.00 p < .001, R2 = .16, and only block number found to be a predictor of reaction time, β = -.385, t = -8.94, p < .001, with a reduction in RTs in relation to task block number. (Fig 4).

A simple linear regression showed that mean reaction time did not predict no-go accuracy, $F_{(1, 113)} < .001$, p = .970, $R^2 < .001$.

## Initial state fatigue

We expected the state fatigue change and the energy change induced by the SART to be predicted by the scores on the five MFI subscales. Likewise, we expected a relationship between the MFI subscales and the pre-task VAS-F scores. A multiple linear regression examined the relationship between all 5 of the MFI subscales and pre-task fatigue, and another examined the relationship of the 5 MFI subscales to pre-task energy, see Fig 5.

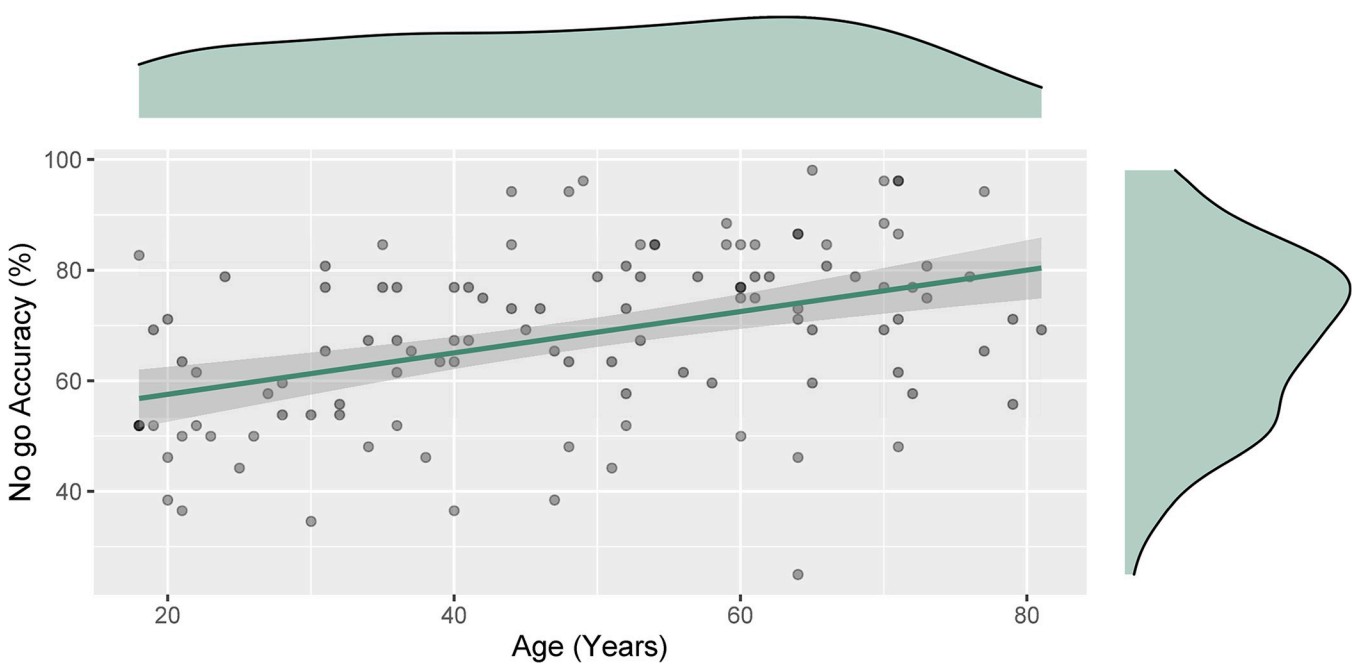

**Fig 3. Linear relationship between participant age and overall participant no-go accuracy with 95% confidence intervals.** Density plots indicating the distribution of the participants across ages 18–81, as well as the no-go accuracy distribution.

The overall model predicting pre-task state fatigue was significant (F(5, 109) = 25.10 p < .001, R2 = .535). Pre-task state fatigue was positively related to mental fatigue (β = .187, t = 2.26, p = .026), physical fatigue (β = .351, t = 3.05, p = .003) and general fatigue (β = .731, t = 6.75, p < .001), but was not related to reduced activity (β = .116, t = 1.10, p = .275) or reduced motivation (β = .088, t = .880, p = .380).

The overall model predicting pre-task state energy was also significant (F(5, 109) = 29.11 p < .001, R2 = .572). Pre-task state energy was negatively related to general fatigue (β = -.672, t = -6.47, p < .001) and reduced activity (β = -.371, t = -3.66, p < .001), whilst showing no relationship to mental fatigue (β = -.090, t = -1.14, p = .256), physical fatigue (β = .211, t = 1.90, p = .060) or reduced motivation (β = .096, t = 1.01, p = .317).

## State change

However, state change only modestly corresponded to the MFI scores. The overall model was significant (F(5, 109) = 2.38 p = .043, R2 = .099). State fatigue change was negatively related to reduced activity (β = -.340, t = -2.31, p = .023), with a small effect size. This meant that participants who had higher reduced activity scores had a smaller increase in their state fatigue during the task. There were no other relationships between state fatigue change and any of the other 4 MFI subscales: mental fatigue (β = -.109, t = -.950, p = .344), physical fatigue (β = -.129, t = -.803, p = .424), general fatigue (β = .098, t = .648, p = .518) or reduced motivation (β = .250, t = 1.80, p = .074). The same model was run to predict state energy change from the five MFI subscales (F(5, 109) = 3.10 p = .012, R2 = .125). Again, reduced activity was also a positive predictor of energy change, β = .377, t = 2.60, p = 0.01 with a small effect size. This showed that participants with higher reduced activity scores reported less energy loss during the task. No relationship was found with the other scales: mental fatigue (β = .046, t = .403, p = .688), physical fatigue (β = .131, t = .829, p = .409), general fatigue (β = -.051, t = -.341, p = .734) and reduced motivation (β = -.212, t = -1.55, p = .124). Therefore, the direction of the relationship

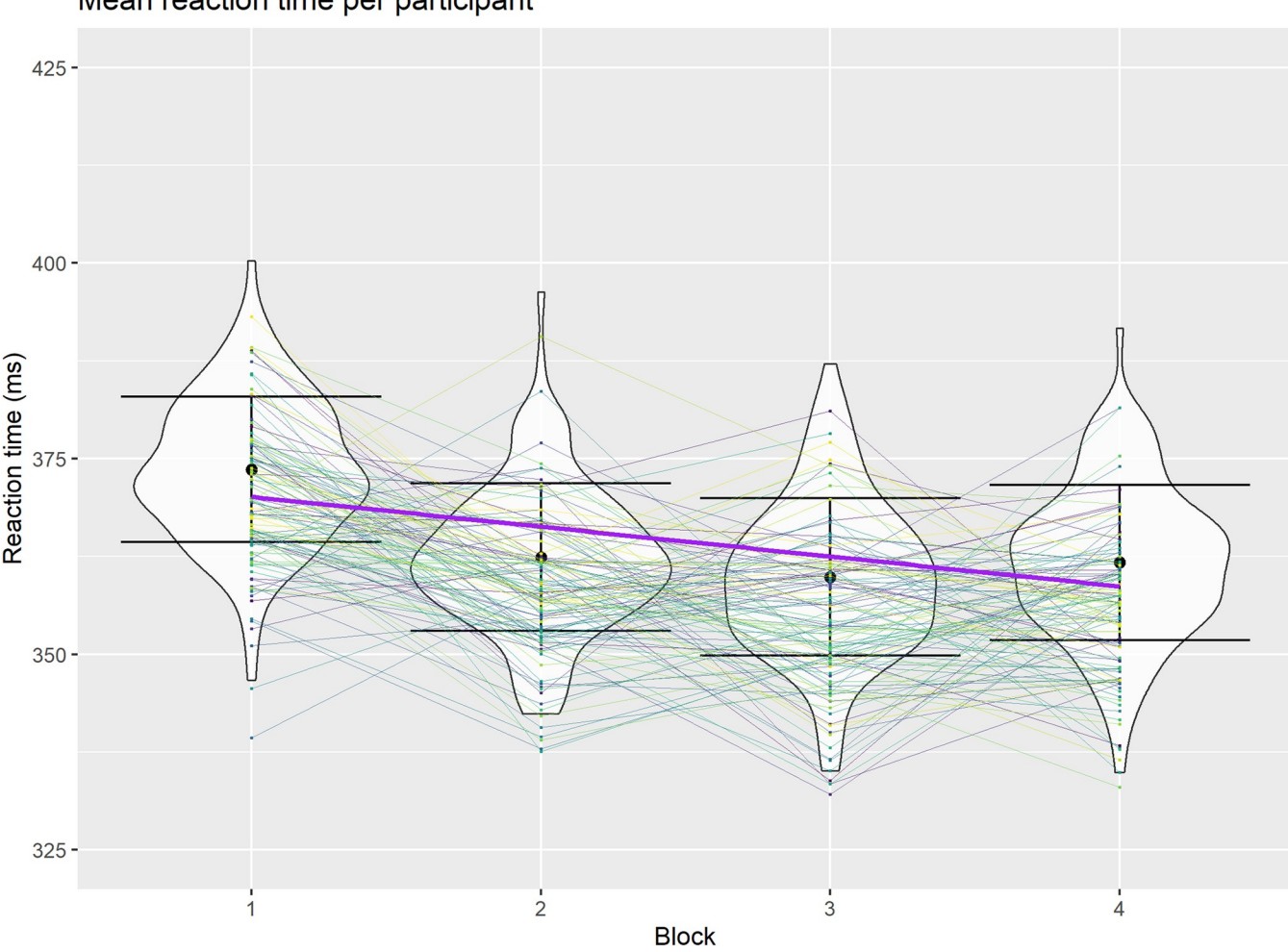

**Fig 4. Reaction times on the SART across time.** Distribution of reaction times at trial block level across the four blocks including mean reaction times and standard deviations with reaction times within each participant across the four task blocks distinguished by colour (including the overall linear trend across the four blocks).

was the opposite of our pre-registered prediction (we predicted that we would observe more task-induced fatigue and energy loss in participants who started the experiment with high levels of reduced activity) (see Fig 6).

All the European sample (86.96%) completed the study in the morning, contrary to the pre-registration, so no analysis was conducted on the difference between morning and evening due to the high imbalance between the two conditions. Likewise, too few participants identified as having experienced Covid symptoms 3 months prior to their participation (4.35%), and so Covid was not considered in the analysis. Employment was too diversified to be categorised for the analysis and so it was not considered either.

## Discussion

### Fatigue and SART

Participants reported fatigue and energy levels before and after carrying out an online version of SART. Levels of state fatigue were predicted by scores from the MFI, which was used to

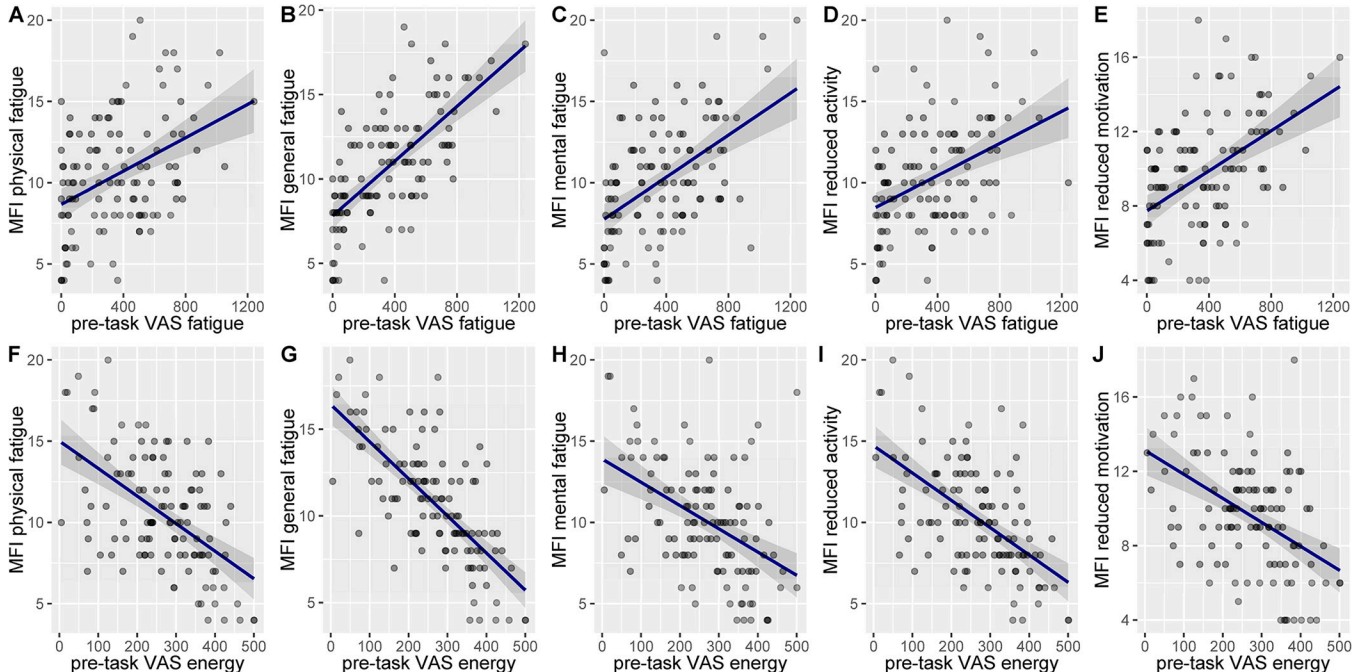

**Fig 5. Trait-state fatigue relationship (including highlighted 95% confidence intervals) on all MFI and pre-task VAS subscales.** (A) VAS fatigue and MFI physical fatigue. (B) VAS fatigue and MFI general fatigue. (C) VAS fatigue and MFI mental fatigue. (D) VAS fatigue and MFI reduced fatigue. (E) VAS fatigue and MFI reduced motivation. (F) VAS energy and MFI physical fatigue. (G) VAS energy and MFI general fatigue. (H) VAS energy and MFI mental fatigue. (I) VAS energy and MFI reduced activity. (J) VAS energy and MFI reduced motivation.

reflect trait fatigue, illustrating that pre-task state fatigue levels were initially rooted in long-term internal trait fatigue. In addition, the findings show that undergoing the SART does induce changes in state fatigue. Participants got fatigued and lost energy during the task, as demonstrated by the clear difference between the pre- and post-task subjective fatigue and energy levels. Furthermore, the change was reflected in their task performance: If performance on the task dropped, more fatigue and less energy were experienced. So, the objective measure of accuracy change between the first and final blocks corresponded to the change in reported subjective fatigue. In fact, state fatigue change as a predictor of accuracy change outperformed initial trait fatigue levels. This was true even though participants received no feedback and were not given any expectations about the desired performance. Our data thus demonstrate a tight coupling between drops in SART accuracy and changes in state fatigue.

We detected an unexpected relationship between state fatigue change and trait fatigue opposite to our original prediction. Change in state fatigue was smaller with higher levels of trait physical fatigue and reduced activity present before the task. This may indicate a ceiling effect, or limited capacity for fatigue change, if a relatively higher fatigue state is already present: when fatigue is high prior to the start of the task, no further increase may be possible given the nature of the task. Our selection of the SART to induce fatigue was motivated by its ability to induce errors in the withholding of a behavioural response caused by lapses in sustained attention. The prolonged, and repetitive nature of this task may have induced a different type (or a different degree) of fatigue compared to a task that, for example, depletes cognitive resources due to high complexity [7, 73] or involves high working memory load [74]. It may be the case that participants with high trait fatigue did not experience greater state fatigue changes because they had a higher tolerance for unstimulating, repetitive tasks, and it is possible that the use of a challenging task may have produced different results in this regard, which

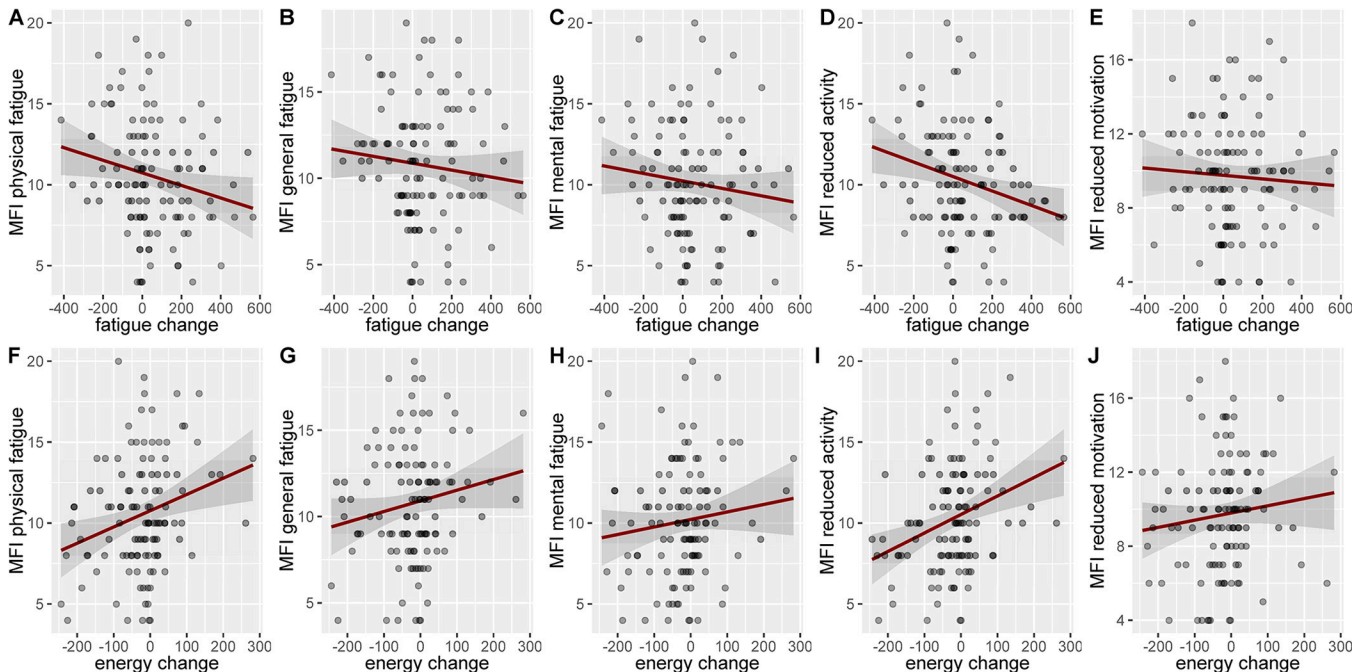

**Fig 6. The fatigue and energy change state and trait fatigue relationship in the sample with highlighted 95% confidence intervals.** (A) VAS fatigue change and MFI physical fatigue. (B) VAS fatigue change and MFI general fatigue. (C) VAS fatigue change and MFI mental fatigue. (D) VAS fatigue change and MFI reduced fatigue. (E) VAS fatigue change and MFI reduced motivation. (F) VAS energy change and MFI physical fatigue. (G) VAS energy change and MFI general fatigue. (H) VAS energy change and MFI mental fatigue. (I) VAS energy change and MFI reduced activity. (J) VAS energy change and MFI reduced motivation.

should be investigated in future research. The findings parallel those of Boolani and colleagues [75], who likewise detected that higher trait fatigue in individuals meant the inability to get further fatigued. The effect has also been suggested to occur due to depleted cognitive reserve on part of the participants, preventing any further rises in fatigue [76].

## Age and learning effects

Age proved to be the strongest predictor of overall performance over any other measures or variables (the older the participants the greater the accuracy), confirming and reproducing the outcomes of a recent meta-analysis by Vallesi and colleagues [34]. We propose this finding to be a robust indicator of an underlying age-specific difference, likely the cognitive approach to the task (see further elaboration under 'future directions').

In agreement with prior findings [36, 37], participants in this large sample sped up over time. There was no speed-accuracy trade off, accuracy remained the same whilst reaction times reduced. This improvement in speed thus showed a gradual increased familiarity with the task and accommodation to the experimental paradigm [18]. Interestingly, this effect occurred regardless of age. It highlights the need to track performance across a larger time window (ideally 40 minutes or longer [18]), should future attempts aim to achieve a measurable decline in SART participant performance.

## Online environment

The achieved go and no-go accuracy rates as well as reaction times when adjusting for errors caused by performing the study online [52] matched those in other studies carried out in the

general population [58, 77]. Notably, the described online performance appears comparable with samples in the laboratory studies [32, 58]. Thus, the findings support the notion that online behavioural research can closely match the laboratory setting, whilst reaching diverse participant groups, which would be harder to recruit in laboratory studies. They also demonstrate the feasibility of obtaining decrement effects despite reduced experimental control over participants than in laboratory settings, such as less influence on timing of breaks between blocks. To our knowledge, this is the first time that the SART has been implemented online, and it achieved very comparable data to laboratory settings. The study thus successfully shows the suitability of such efforts for future attempts to investigate ageing as well as the link between subjective measures of fatigue and SART performance change. It also provides a characteristic profile of fatigue and baseline fatigue rates prior to experimentation in online samples across the whole adult lifespan with the exception of advanced age, as well as the influence of fatigue levels over the course of the experiment.

## Limitations

The original pre-registered models were based on concrete linear predictions between one predicted and one predictor variable, but we considered several variables within one model, not all of which were directly mentioned in the pre-registration. Nonetheless, the models broadly reflect the anticipated pre-registered relationships. This approach helped us to detect the unanticipated link between increased trait fatigue and reduced state fatigue change. However, this remains an initial exploratory finding and does not exhaust the full potential of the available dataset. Other analytical approaches were not considered to further explore this correspondence of trait to state fatigue, including a promising use of non-linear models already utilised elsewhere [78].

We are aware that other factors could have impacted state fatigue and there is added uncertainty in the use of an online environment, which is usually alleviated by more precise laboratory control over the experiment: factors pertaining to the sleep cycle and time of day which could impact the vigilance decrement [79] were not considered, and other insufficiently addressed factors include individual problems in nourishment [4], gender, work exhaustion, smoking or undetected underlying health conditions which could in turn have contributed to the reported trait and state levels of fatigue. Furthermore, older participants in the present sample may have different motivational levels, an effect previously seen in laboratory samples [80]. Future replications should aim to survey motives for participation and rates of mind-wandering to isolate potential age-related confounds.

## Future directions

It was interesting to see that age had a greater impact on overall SART performance than initial fatigue levels, yet this may be the case only for a generally healthy population (as tested here). Research into clinically fatigued populations is warranted to see whether fatigue becomes a significant predictor of SART performance, as the SART may be much more challenging for patients with diagnosed clinical fatigue conditions. This study highlights the significance of SART as an objective measure of fatigue change, and it may well prove to be a sensitive, objective means of assessing and monitoring fatigue in clinically fatigued populations.

Reverting back to the age effect, the present findings provide clear evidence of a stable age effect in the standard implementation of the SART. The used sample size allowed us to treat age as a continuous variable and so show a linear relationship between higher accuracy and age. There are clear differences which occur in the attentional processes necessary to undertake this task with increasing age. One existing explanation is that the task is either perceived as

more interesting and challenging, and so carried out more dutifully with advancing age. Thus, older participant performance hints at a motivational advantage [34] and reliance on a more accuracy-based cognitive strategy [39]. At the same time, it may paradoxically show that improved performance means greater difficulty and necessity to actively engage cognitive resources when performing the task. The task could be perceived as more routine and automatic by younger participants due to its relative simplicity [34]. They would therefore opt for a speed-based strategy. In contrast, older adults may experience a more innate motivational drive to excel at the task [77, 78]. Future work could investigate this phenomenon in more detail by utilising motivational manipulations and probing attitudes to experimentation in participants. Pairing future studies of the SART with neural measures tracking the employment of cognitive resources independent from subjective report would enable the detection of this speculated effect. Recent research has started to investigate oscillatory neural correlates of performance on the SART [39, 81] focusing on the link between the SART and brain oscillations in particular [49, 81, 82]. Further research could link changes in the oscillatory signal to this objective performance and the subjective experience of fatigue, and this would help to ground research on fatigue in clinically relevant theoretical conceptualisations [29, 83] as well as help to better understand innate and lasting proclivity to fatigue. Another open avenue of research is the comparison of this attentional approach to other paradigms and tasks with higher cognitive demands [84, 85] or using a task-switching approach [86].

## Conclusion

In summary, we investigated the impact of undergoing the SART in a large online sample comprising all adult age groups. We found that an increase in reported state fatigue was reflected in reduced SART performance. We also found that age, not trait predisposition to fatigue, was the greatest predictor of overall performance on the task. Pre-task trait fatigue led to a ceiling effect in state fatigue change only. We propose that the SART is a sensitive, objective means to induce and measure changes in state fatigue.

## Supporting information

**S1 File. Outline of subjective measures.**
(DOCX)

**S1 Appendix. Revised linear model tables.**
(DOCX)

## Acknowledgments

We would like to extend our gratitude to all the reviewers and the editors who contributed to the finalisation of this article.

## Author Contributions

**Conceptualization:** Gemma Learmonth, Gregor Thut, Monika Harvey.

**Formal analysis:** Simon Hanzal.

**Investigation:** Simon Hanzal.

**Methodology:** Simon Hanzal, Gemma Learmonth.

**Software:** Simon Hanzal.

**Visualization:** Simon Hanzal.

**Writing – original draft:** Simon Hanzal.

**Writing – review & editing:** Gemma Learmonth, Gregor Thut, Monika Harvey.

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
