## [Decision Letter · Decision Letter 0]

27 Dec 2023

PONE-D-23-30888Probing sustained attention and fatigue across the lifespanPLOS ONE

Dear Dr. Hanzal,

Thank you for submitting your manuscript to PLOS ONE. After careful consideration, we feel that it has merit but does not fully meet PLOS ONE’s publication criteria as it currently stands. Therefore, we invite you to submit a revised version of the manuscript that addresses the points raised during the review process. Both reviewers agree the manuscript is of interest, and make various suggestions that should improve clarity throughout.

We look forward to receiving your revised manuscript.

Kind regards,

Jessica Adrienne Grahn

Academic Editor

PLOS ONE

“We would like to extend our gratitude to all the reviewers and the editors who contributed to the finalisation of this article. The funders had no role in study design, data collection and analysis, decision to publish, or preparation of the manuscript.”

“SH was supported by the Economic and Social Research Council Grant: ES/P000681/1. https://www.ukri.org/councils/esrc/

GM was funded by the Wellcome grant 209209/Z/17/Z.

https://wellcome.org/

Reviewers' comments:

Reviewer's Responses to Questions

**Comments to the Author**

1. Is the manuscript technically sound, and do the data support the conclusions?

Reviewer #1: Partly

Reviewer #2: Partly

2. Has the statistical analysis been performed appropriately and rigorously? 

Reviewer #1: No

Reviewer #2: No

3. Have the authors made all data underlying the findings in their manuscript fully available?

Reviewer #1: Yes

Reviewer #2: Yes

4. Is the manuscript presented in an intelligible fashion and written in standard English?

Reviewer #1: Yes

Reviewer #2: Yes

5. Review Comments to the Author

Reviewer #1: The manuscript investigated the association of fatigue (at the trait and state level) and sustained attention task performance across a sample of individuals from young to old age. Performing the attention task resulted in greater self-reported state fatigue, and state fatigue was predicted by initial pre-task trait fatigue ratings. Age was also associated with task performance measures. There were also some unexpected findings, such as a negative relationship between change in state fatigue (from before to after performing the task) and trait fatigue. The study also served as a proof of concept for the administration of the SART in an online experiment. Overall, the study is interesting and addresses an important issue – linking the subjective sense of fatigue and individual experiences with objective performance measures. My primary concerns with the paper centered on a few areas: (1) the use of a sustained attention task rather than a cognitively demanding task may induce fatigue in different ways, and produce different results – this distinction can be further clarified both in the introduction and discussion; (2) the use of a stepwise regression approach may produce results that are somewhat idiosyncratic, and may obscure the joint influence of multiple predictors in the models; and (3) there are some elements of the online administration procedure that are not entirely clear, which may influence the findings, such as the monitoring of the timing of breaks for study blocks. More specific comments related to these concerns and others are noted further below:

Introduction:

-I am a bit confused by the sentence: “We did not propose trait fatigue to affect task performance directly, but trait fatigue to affect state fatigue and thus indirectly contribute to changes in performance, and changes in state.” If certain measures of task performance are reflective of state fatigue, and trait fatigue directly influences state fatigue, then would this not be a direct affect on task performance? Perhaps this can be clarified a bit further.

-the distinction and/or similarity between “fatigue” and “energy” could be further outlined.

-Trait fatigue may also not only be defined as fatigue over a prolonged period of time, but rather it is often further characterized as an individual’s inherent tendency to experience fatigue across a range of situations. Perhaps this is not so different than the way it is defined in the text, but it may be useful to integrate this latter conceptualization of trait fatigue further into the introduction. I ask that the authors consider if this latter conceptualization provides some unique perspective to the way it is already presented in the paper.

-Specific hypotheses associated with each aim could be added to the paper.

Methods:

-was the wording adjusted for some items of the VAS-F due to regional language norms? The reliability of the scales is presumably based on the original wording used in the scales, so further comment on this adjustment is warranted.

-Were there any other specific inclusion/exclusion criteria for participants? For example, normal vision, no history of cognitive/neurological impairment, etc. This should also be noted in the participants section of the methods.

-lines 216-218: I am not following the logic of the “at chance level” phrasing here. I understand the general reasoning, which is that people could adopt a strategy where they just tend to either respond or not respond regardless of the stimulus – but I am not quite understanding how this is at chance level (like flipping a coin). These strategies would bias the response either towards higher overall accuracy (at the expense of nogo inaccuracy) or lower overall accuracy (to the benefit of nogo accuracy). Is “at chance level” the correct wording to use here?

-It is unclear to me why the additional 10ms is expected for response times in the online task in comparison to in the laboratory. The expectation of increased variability seems understandable, as various systems will have different precision for detecting response speed over multiple trials. But is there a reason that a systematic delay would be expected as well, rather than just increased variability, across participants’ computers?

-Was the number sequence randomized for each individual participant, or was the sequence of numbers the same for each person?

-It appears that a stepwise regression model was used to identify predictors that would remain in the model based on whether they reached a specific significance threshold. In general, stepwise regression can result in idiosyncratic findings that may be difficult to replicate across studies, because they increase the number of tests that are being performed and are susceptible to enhancing the impact of false positives that may be present in a given dataset. In contrast, a hypothesis driven regression model that includes a pre-specified, well-reasoned set of predictor variables is thought to produce more reliable findings. The predictors that the authors identify seem reasonable, and I suggest that the model should just include all of these predictors, especially if the authors believe that each of these predictors is a potential contributor to variance in the outcome measure. The sample size is certainly large enough for the number of predictors (I believe five in total), and the information gained from this analysis design would be more informative and potentially more reliable than a stepwise regression approach.

-The effects of fatigue on the SART are being estimated by comparing performance between the first and fourth blocks. However, on the fourth block, participants are still starting the block after having a break. So it is somewhat unclear how much performance should be expected to decline if there are sufficient breaks between each block.

Furthermore along these lines: was there any data collected regarding the length of the breaks that participants took between blocks? Did participants take different lengths of breaks, on average, based on the age stratification? This is a potential confound in the data that should be reported, and which could also potentially be further controlled for in the statistical analyses.

Results:

-What was the reason for log-transforming the reaction time data? Were the reaction time data skewed? If they were normally distributed, then why is the log transformation necessary?

-A table should be included that provides a full overview of the relevant statistics for the regression models.

-The left panel in figure 4 appears to show identical mean reaction time and SD of RT across the four blocks of the task. However, the trendline in the right panel, and the analyses reported in the text, both appear to indicate that there is a reduction of RT overall across the blocks. It seems that there is a discrepancy between these figures, or is there some other additional information I am missing here? A similar question applies regarding the variability (SD) around the mean.

-Were corrections for multiple comparisons applied to other analyses that did not include age and gender? If that is not the case, then is unclear why the correction was applied only when these factors were present in the analyses.

Discussion:

-I am curious if the improving performance on the SART as a function of increasing age is influenced in some way by the level of fatigue an individual experiences (either trait or state). Do individuals with greater reported fatigue not show the same level of improvement across task blocks? The discussion on page 25 suggests that the older participants may have been more interested in the task – but I wonder if this interest would wane for individuals that are highly fatigued, and if this effect would be stronger in older compared to younger participants.

-The ability to sustain attention (as represented in the SART) could be conceptualized as overlapping with, but also distinct from, fatigue in different ways. Cognitive fatigue, in particular, is thought to be induced when people have to engage in a cognitively challenging task with high demands. In contrast, the SART is not necessarily cognitively challenging to the same degree as some other tasks. So the findings of the study could be different based on whether one is measuring sustained attention vs. a cognitively demanding task, such as for example a challenging dual-task paradigm or task with high working memory or executive demand. This nuance of the findings could be further outlined in the discussion (and maybe it would be worth addressing this further in the introduction as well).

Other minor comments:

Abstract: perhaps some additional text could be added to provide further context for the meaning of “reduced energy” in the abstract.

-line 38, typo? – “the3”

-line 56: MFI should be defined at first use.

-line 89: check phrasing of the sentence.

-line 124: should just say “SART” without “task” afterwards, I believe, as the “task” is redundant with the “T” in the acronym.

-I am unclear what is meant by the phrase “a balanced mean” on page 16.

-line 476, typo: “reaction times”.

-line 480, phrasing: “the laboratory setting”

Reviewer #2: Overall I think this study is a very good study however, this study ignores much of the published works of O'Connor, Boolani and Filippi. The work of Fillipi (Fillipi, et al, 2022) suggests that what you are measuring with MFI may be prolonged state fatigue and not necessarily trait fatigue. Further, works by both Boolani and O'Connor suggest that trait fatigue is the long-standing pre-disposition to fatigue and not how someone has felt lately, as the MFI instructions suggest.

Further, recent work by Boolani (>5 manuscripts) support that trait mental and physical energy and fatigue modify state energy and fatigue in the direction that your results suggest (high trait fatigue results in a lower increase in state fatigue than low trait). This may be due to the fact that those who normally feel fatigued (high trait fatigue) do not have much reserve left and are already starting more fatigued than their counterparts so they can't possibly feel worse. Boolani and colleagues have noted these changes in multiple studies. Their control interventions may provide you with comparisons.

Another issue with this work is that the authors do not acknowledge that energy and fatigue are two distinct uniolar moods with their own distinct biological, behavioral and biomechanical correlates.

Secondly, I downloaded your data and did some visualizations and ran Shapiro-Wilks tests for normality and your pre and post VAS scales were not normally distributed (after calculating for fatigue and energy scores). I ran the same for deltas and saw that while fatigue was normally distributed, energy was not. Did you apply any sort of data transformation techniques to these scores? Same with age, you had more people in the late teens/early 20s in the 18-29 cohort, the same goes for the 70-81.

I think the statistics need a closer look as distribution of data may impact your models.

6. PLOS authors have the option to publish the peer review history of their article (what does this mean?). If published, this will include your full peer review and any attached files.

Reviewer #1: No

Reviewer #2: No

---

## [Author Response · Author response to Decision Letter 0]

9 Feb 2024

Dear Dr Grahn,

Many thanks to you and reviewers for dedicating their time to reviewing our manuscript and offering most valuable constructive suggestions. In response to the feedback received, we have carefully revised our manuscript, incorporating all the provided comments. The following section comprehensively addresses each comment, detailing the steps we have taken to incorporate them. Changes made to the initial text are clearly highlighted in the revised manuscript. Below we provide responses to the comments and actions that we took to address them. The responses are highlighted in blue in the accompanying response to reviewers file.

Yours sincerely,

Simon Hanzal, Gemma Learmonth, Gregor Thut and Monika Harvey

Authors of PONE-D-23-30888

Probing sustained attention and fatigue across the lifespan

Detailed Response:

• included

• included

• included

Authors’ response: We have revised the manuscript to follow the PLOS ONE template. 

“We would like to extend our gratitude to all the reviewers and the editors who contributed to the finalisation of this article. The funders had no role in study design, data collection and analysis, decision to publish, or preparation of the manuscript.”

“SH was supported by the Economic and Social Research Council Grant: ES/P000681/1. https://www.ukri.org/councils/esrc/

GL was funded by the Wellcome Trust 209209/Z/17/Z.

https://wellcome.org/

Authors’ response: We have removed the duplicated text from the manuscript, which now only reads (page 26):

“We would like to extend our gratitude to all the reviewers and the editors who contributed to the finalisation of this article.”

Authors’ response: We have added the following ethics statement at the end of the participant section (page 14): 

“The study was approved by the University of Glasgow College of Science and Engineering Ethics committee (Approval number: 300200069). Digital consent was obtained from all participants by completing the online checkbox form."

Reviewer 1: 

The manuscript investigated the association of fatigue (at the trait and state level) and sustained attention task performance across a sample of individuals from young to old age. Performing the attention task resulted in greater self-reported state fatigue, and state fatigue was predicted by initial pre-task trait fatigue ratings. Age was also associated with task performance measures. There were also some unexpected findings, such as a negative relationship between change in state fatigue (from before to after performing the task) and trait fatigue. The study also served as a proof of concept for the administration of the SART in an online experiment. Overall, the study is interesting and addresses an important issue – linking the subjective sense of fatigue and individual experiences with objective performance measures. 

Authors’ response: We thank the reviewer for this positive assessment of our study.

My primary concerns with the paper centered on a few areas: (1) the use of a sustained attention task rather than a cognitively demanding task may induce fatigue in different ways, and produce different results – this distinction can be further clarified both in the introduction and discussion; 

Authors’ response: We thank the reviewer for this valid point and detail our response further below.

(2) the use of a stepwise regression approach may produce results that are somewhat idiosyncratic, and may obscure the joint influence of multiple predictors in the models; 

Authors’ response: Again, please see detailed response further below.

and (3) there are some elements of the online administration procedure that are not entirely clear, which may influence the findings, such as the monitoring of the timing of breaks for study blocks. 

Authors’ response: Again, please see detailed response further below.

More specific comments related to these concerns and others are noted further below:

Introduction:

-I am a bit confused by the sentence: “We did not propose trait fatigue to affect task performance directly, but trait fatigue to affect state fatigue and thus indirectly contribute to changes in performance, and changes in state.” If certain measures of task performance are reflective of state fatigue, and trait fatigue directly influences state fatigue, then would this not be a direct affect on task performance? Perhaps this can be clarified a bit further.

Authors’ response: Thank you for pointing this out. Our main reason to believe that state fatigue would have a greater impact on performance than trait fatigue is its greater sensitivity to task-related changes. While a long-term predisposition to fatigue may be among predictors of performance, the actual state the participants are in was seen as the main indicator of performance. However, we do propose that state fatigue could be rooted in trait fatigue. We have further clarified this in the text, where it now reads (Introduction, page 6): 

“We hypothesised that trait fatigue could either negatively affect task performance directly, or that trait fatigue may predispose participants to higher levels of pre-task state fatigue. In turn, we proposed that this pre-task state fatigue may cause further in-task changes to state fatigue, and consequently to performance on the SART.”

-the distinction and/or similarity between “fatigue” and “energy” could be further outlined.

Authors’ response: Thank you for the suggestion. Originally, we chose to use the Lee et al., (1991) VAS measure for its multi-item nature and to assess the direct momentary experience of fatigue, in preference to the single-item measure of state more typically used (Seli et al., 2013, Wylie et al., 2022). The choice of this measure with two specific subscales for “fatigue” and “energy” (and hence the choice of terminology), was thus not theory driven. Instead, wherever the original pre-registration referred to state fatigue, it denoted both subscales used in the VAS measure, the fatigue (bearing the same name) and energy subscales. However, we now distinguish between fatigue and energy in more detail in the text (Introduction, page 4):

“There has been a tight coupling of state fatigue with energy, where several studies (13,24, 25) utilised combined measures of state fatigue comprised of energy and fatigue subscales. Although the two scales seem to be closely related, findings of divergent changes in both suggest that they constitute two related, unipolar aspects (13). Accordingly, a more in-depth investigation of state fatigue benefits from inclusion of both of the two separate subscales.”

-Trait fatigue may also not only be defined as fatigue over a prolonged period of time, but rather it is often further characterized as an individual’s inherent tendency to experience fatigue across a range of situations. Perhaps this is not so different than the way it is defined in the text, but it may be useful to integrate this latter conceptualization of trait fatigue further into the introduction. I ask that the authors consider if this latter conceptualization provides some unique perspective to the way it is already presented in the paper.

Authors’ response: Thank you for this insightful observation. Indeed, we do not see such additional specification of trait fatigue in any way running contrary to our arguments, but as a beneficial and interesting theoretical point to elaborate. We have thus nuanced our description of MFI capturing fatigue as a trait in the introduction with the addition of the following (Introduction, page 3):

“It has been characterised as an innate tendency to exhibit fatigue (13)”.

And further on: “This type of recent, self-reported fatigue has also been described as “prolonged state fatigue” (Filippi et al., 2022) in order to distinguish it from an innate tendency to become fatigued (sometimes also described as “trait fatigue”).”

And further on in the discussion (page 25): “as well as help to better understand innate and lasting tendency towards fatigue.”

-Specific hypotheses associated with each aim could be added to the paper.

Authors’ response: Thank you for this consideration. We have now listed the specific hypotheses under each of the aims at the end of the introduction section (page 7), as follows: 

1) Investigate the correlational relationship between changes in state fatigue and performance changes on SART over time. Specifically, that no-go accuracy will decrease, and reaction times will increase as a) state fatigue increases, and b) state energy decreases. 

2) Assess the relationship between no-go trial accuracy and reaction time on the SART and reported trait and state fatigue. Specifically, that no-go accuracy will be lower, and reaction times will be slower in participants with a) high trait fatigue, b) high state fatigue and c) low state energy.

3) Determine the relationship between subjective trait fatigue and state fatigue, as well as changes in state fatigue as a result of the task. Specifically, that trait fatigue would be a) positively correlated with pre-task state fatigue and b) negatively correlated with pre-task state energy. We also expected to observe greater SART-induced c) increase in state fatigue when pre-task trait fatigue was high, and d) decrease in state energy when pre-task trait fatigue was high. 

4) Carry out the research project online, targeting the general population across the whole lifespan and so test the viability of an online environment for general research on sustained attention. Based on previous divergent findings, we expected to observe difference in a) no-go accuracy and b) reaction times with increasing age.

Methods:

-was the wording adjusted for some items of the VAS-F due to regional language norms? The reliability of the scales is presumably based on the original wording used in the scales, so further comment on this adjustment is warranted.

Authors’ response: Thank you for this valid point: We have now expanded the text to point out this change and to include the effect on reliability (Materials and methods, page 9): 

“Two items of outdated language were replaced by closest possible synonyms: “worn out” to “drained”, and “bushed” to “run down” to avoid a poor understanding of the items. Subsequent internal consistency tests confirmed no impact on the measure’s reliability.”

-Were there any other specific inclusion/exclusion criteria for participants? For example, normal vision, no history of cognitive/neurological impairment, etc. This should also be noted in the participants section of the methods.

Authors’ response: Apologies for this omission. We amended the text to read (Materials and methods, page 8): 

“Participants were admitted to the experiment if they indicated at least moderate proficiency in the English language and reported an absence of any cognitive or neurological conditions or uncorrected vision.”

-lines 216-218: I am not following the logic of the “at chance level” phrasing here. I understand the general reasoning, which is that people could adopt a strategy where they just tend to either respond or not respond regardless of the stimulus – but I am not quite understanding how this is at chance level (like flipping a coin). These strategies would bias the response either towards higher overall accuracy (at the expense of nogo inaccuracy) or lower overall accuracy (to the benefit of nogo accuracy). Is “at chance level” the correct wording to use here?

Authors’ response: Thank you for highlighting this ambiguity. We acknowledge that the use of the word “strategy” to define the expected, deliberately erroneous, approaches that participants might take to the task was unfortunate, leading to confusion with cases where participants used valid strategies to carry out the task correctly. The text has now been amended (Materials and methods, pages 11-12): 

“As per the pre-registered protocol, participants with responses that fell into either of two anticipated deliberately erroneous approaches to completing the experiment were removed from further analysis: One was responding to all trials and withholding responses randomly at the rate of the occurrence of the no-go stimuli (> 89% go stimuli correct and < 11% no-go stimuli correct), the other to randomly respond at the rate of occurrence of the no-go stimuli (< 11% go stimuli correct and >89% no-go stimuli correct), in any of the four experimental blocks.”

-It is unclear to me why the additional 10ms is expected for response times in the online task in comparison to in the laboratory. The expectation of increased variability seems understandable, as various systems will have different precision for detecting response speed over multiple trials. But is there a reason that a systematic delay would be expected as well, rather than just increased variability, across participants’ computers?

Authors’ response: Thank you for making this point. Our expectation was rooted in the findings of Bridges and colleagues (Bridges et al., 2020), who maintain that participants tend to access online experiments using devices with lower screen refresh rates. This can lead to a more imprecise display of stimuli in relation to participant reaction times in comparison with laboratory studies. The paper also considers lag as a factor in online experimentation. Since we have not directly compared the two, we decided to simplify the discussion to focus on variability. We have amended the text to read (Materials and methods, pages 10): 

“We expected a greater standard deviation in response times in our online sample in comparison to the conventional experiments, due to possible lower accuracy and variability of the devices the participants could use to access the task (49).”

-Was the number sequence randomized for each individual participant, or was the sequence of numbers the same for each person?

Authors’ response: Thank you for highlighting this matter. We have added more detail information into the method section clarifying this (Materials and methods, page 10): 

“For each participant, the numbers were sampled randomly, but with each number appearing the same number of times.”

-It appears that a 

---

## [Decision Letter · Decision Letter 1]

19 Mar 2024

PONE-D-23-30888R1Probing sustained attention and fatigue across the lifespanPLOS ONE

Dear Dr. Hanzal,

Thank you for submitting your manuscript to PLOS ONE. After careful consideration, we feel that it has merit but does not fully meet PLOS ONE’s publication criteria as it currently stands. Therefore, we invite you to submit a revised version of the manuscript that addresses the points raised during the review process.

There are some minor comments suggested by one reviewer that I do think would be relevant to address, and I would anticipate no need to send out for review again.

We look forward to receiving your revised manuscript.

Kind regards,

Jessica Adrienne Grahn

Academic Editor

PLOS ONE

Journal Requirements:

Reviewers' comments:

Reviewer's Responses to Questions

**Comments to the Author**

1. If the authors have adequately addressed your comments raised in a previous round of review and you feel that this manuscript is now acceptable for publication, you may indicate that here to bypass the “Comments to the Author” section, enter your conflict of interest statement in the “Confidential to Editor” section, and submit your "Accept" recommendation.

Reviewer #1: (No Response)

2. Is the manuscript technically sound, and do the data support the conclusions?

Reviewer #1: Yes

3. Has the statistical analysis been performed appropriately and rigorously? 

Reviewer #1: Yes

4. Have the authors made all data underlying the findings in their manuscript fully available?

Reviewer #1: Yes

5. Is the manuscript presented in an intelligible fashion and written in standard English?

Reviewer #1: Yes

6. Review Comments to the Author

Reviewer #1: The authors have made substantial revisions to the paper, and I believe it is much improved. I have provided some additional comments below that the authors could further consider:

Abstract:

-There should be a reference for the quoted text in the first sentence, if it comes directly from a source.

-Could just say “An online version” and leave the JsPsych part out of the third sentence – as this could potentially lead to some confusion about what this is referring to without further context. It seems unnecessary to note the software used to administer the task in the abstract, from my perspective.

Introduction:

-There could be a brief description of the way that the SART is administered when this is first mentioned.

-The conceptual distinction between energy and fatigue could still be further described in the introduction.

-The “Online Research” section feels a bit out of place. I wonder if this could possibly be combined with another section in a way that makes it less abrupt when reading the introduction. Just something to consider.

-suggested edit for the aims: “Specifically, we predict that no-go accuracy…”

Methods

-What is the relevance of the statement: “All European participants completed the study in the morning…” Why was this done? Also, did non-European participants complete the study at a different time? This is mentioned again later, but I am wondering why there is a distinction in here on the European vs. non-European participants.

Results:

-It is more conventional to include the statement on informed consent in the methods section, rather than the results.

-In the text (e.g., page 16, line 341), a beta value of < .001 is noted for the fatigue change predictor. However, inspection of the supplementary tables shows that this is the unstandardized coefficient. There are other examples of this elsewhere in the results section. I suggest that the standardized beta coefficients can also be added to the tables, to provide further context for the size of the effects regardless of measurement tool scale in the regression analyses. Given the very small values for the unstandardized coefficients, I think it could be helpful to also include the standardized coefficients.

Discussion:

-I wonder about another potential explanation for the finding that older participants had better nogo performance than younger participants; and that greater fatigue change was associated with poorer nogo performance. Does this mean also that younger people had greater fatigue change with time on task? That is perhaps also somewhat unexpected. Another potential factor that should be considered in the discussion is how the at-home, online nature of the task may have resulted in these findings. The use of the remote delivery of the test is discussed, but primarily in the context of its usefulness for engaging in research that uses an online delivery method. However, this approach could also have a different impact on fatigue and performance in younger vs. older individuals, possibly due to novelty of the experience. Perhaps, younger individuals, who may potentially engage in online activities more frequently and proficiently in some regard, may also engage in multitasking more frequently when engaging in analogous activities. Therefore, maintaining attention towards a single, relatively simple task of this type may lead to greater distractibility and be more mentally draining for younger individuals (maybe they were even multitasking a bit – was this possible or monitored?). In contrast, older individuals may find this experience to be more novel (and maybe more “formal”), and therefore may not be as distractable or find the task as mentally draining to engage with for a prolonged period of time. This speculation leads me to wonder if the results would be replicated in the in-lab setting. What contexts might change these findings? It could be worth further considering this and adding to the discussion along these lines.

7. PLOS authors have the option to publish the peer review history of their article (what does this mean?). If published, this will include your full peer review and any attached files.

Reviewer #1: No

---

## [Author Response · Author response to Decision Letter 1]

24 Apr 2024

Dear Dr Grahn,

Thank you and the reviewers for generously dedicating your time to carefully reviewing the revised manuscript and identifying further points for improvement. We have revised the manuscript in light of the feedback received, integrating all the comments provided. The subsequent text discusses each, outlining the measures we have implemented to address them. Changes made are clearly highlighted in the revised manuscript. Below, the responses are provided with comments and actions that were taken to address them. The responses are highlighted in blue (pdf version).

Yours sincerely,

Simon Hanzal, Gemma Learmonth, Gregor Thut and Monika Harvey

Authors of PONE-D-23-30888

Probing sustained attention and fatigue across the lifespan

Detailed Response:

• Included (Second Response to Reviewers)

• Included (Second Revised Manuscript)

• Included (Second Manuscript)

Journal Requirements:

Author’s response: The reference list was reviewed for retractions of papers and none were found. Slight errors in reporting the doi were corrected.

Reviewers' comments:

Reviewer's Responses to Questions

Comments to the Author

1. If the authors have adequately addressed your comments raised in a previous round of review and you feel that this manuscript is now acceptable for publication, you may indicate that here to bypass the “Comments to the Author” section, enter your conflict of interest statement in the “Confidential to Editor” section, and submit your "Accept" recommendation.

Reviewer #1: (No Response)

2. Is the manuscript technically sound, and do the data support the conclusions?

Reviewer #1: Yes

3. Has the statistical analysis been performed appropriately and rigorously? 

Reviewer #1: Yes

4. Have the authors made all data underlying the findings in their manuscript fully available?

Reviewer #1: Yes

5. Is the manuscript presented in an intelligible fashion and written in standard English?

Reviewer #1: Yes

6. Review Comments to the Author

Reviewer #1: The authors have made substantial revisions to the paper, and I believe it is much improved. I have provided some additional comments below that the authors could further consider:

Abstract:

-There should be a reference for the quoted text in the first sentence, if it comes directly from a source.

Authors’ response: We thank the reviewer for this observation. As the journal requirement specifies that no in-text referencing should occur in the abstract, the introduction of the term state fatigue was reworded (Abstract, page 2):

“whereas state fatigue is a short-term reaction to intense or prolonged effort.”

-Could just say “An online version” and leave the JsPsych part out of the third sentence – as this could potentially lead to some confusion about what this is referring to without further context. It seems unnecessary to note the software used to administer the task in the abstract, from my perspective.

Authors’ response: We agree with the reviewer about the redundancy of this detail in the abstract. Reference to JsPsych was crossed out (page 2) from the text:

“An online version.”

Reference to JsPsych now remains only in the methods section (page 11).

Introduction:

-There could be a brief description of the way that the SART is administered when this is first mentioned.

Authors’ response: We thank the reviewer for highlighting the need to describe the administration of SART, which has now been included at the first mention of the task (Introduction, page 5):

“The task is characterised by frequent go trials where a response is expected and rare no-go trials which tend to elicit errors arising from attentional lapses. Performance on the SART is thus typically measured in terms of response accuracy with a focus on commission errors (i.e., when erroneous responses are made in no-go trials), reaction times, and standard deviations in reaction times.”

-The conceptual distinction between energy and fatigue could still be further described in the introduction.

Authors’ response: We thank the reviewer for noting this need. More text was added in order to better outline the difference between the two concepts and define energy (Introduction, page 4):

“Although the two scales seem to be closely related, findings of divergent changes in both suggest that they constitute two related, unipolar aspects [13,26]. In relation to fatigue, energy was described as the potential to carry out mental or physical activity [26]. The two measures can often share a similar pattern of response to fatiguing tasks [25], but have also been found to differ, especially in the context of physical activity [26,27].”

-The “Online Research” section feels a bit out of place. I wonder if this could possibly be combined with another section in a way that makes it less abrupt when reading the introduction. Just something to consider.

Authors’ response: We thank the reviewer for the valuable suggestion. The section was sightly shortened and integrated with the aims section (Introduction, page 6):

“We leveraged the online approach in the present investigation and hypothesised that in an online experiment, trait fatigue could either negatively affect task performance directly, or that trait fatigue would predispose participants to higher levels of pre-task state fatigue.”

-suggested edit for the aims: “Specifically, we predict that no-go accuracy…”

Authors’ response: We thank the reviewer for the observation. The text was changed to match the suggested edit (Introduction, page 6).

Methods

-What is the relevance of the statement: “All European participants completed the study in the morning…” Why was this done? Also, did non-European participants complete the study at a different time? This is mentioned again later, but I am wondering why there is a distinction in here on the European vs. non-European participants.

Authors’ response: We thank the reviewer for raising this key point. The original text was referring to participants accessing the study from time zones British Summer Time (BST) and Central European Summer Time (CEST), covering most of the land area of Europe. Since the study was released at 8am BST and completed within two hours, it meant that majority of participants completed the study in the same time period of the day, although the study was released globally. This fact is important as both fatigue and energy may be affected by circadian rhythm (Lerdal et al., 2013). The text was extended to explain the reasoning behind this detail and a distinction was made to specify that “European” in this context refers to time zones (Methods, page 15):

“This also meant that since most of the participants who completed the study accessed it from European countries under either British Summer Time (BST) or Central European Summer Time (CEST), the predominant majority of participants completed the study in the morning.”

Results:

-It is more conventional to include the statement on informed consent in the methods section, rather than the results.

Authors’ response: We agree with the reviewer about the correct location of the statement within the text. It has been transferred from the results (page 15) to the methods (page 10).

-In the text (e.g., page 16, line 341), a beta value of < .001 is noted for the fatigue change predictor. However, inspection of the supplementary tables shows that this is the unstandardized coefficient. There are other examples of this elsewhere in the results section. I suggest that the standardized beta coefficients can also be added to the tables, to provide further context for the size of the effects regardless of measurement tool scale in the regression analyses. Given the very small values for the unstandardized coefficients, I think it could be helpful to also include the standardized coefficients.

Authors’ response: We thank the reviewer for pointing out the importance of including fuller information on the re-run models in the supplementary tables. We agree that the small values, although not incorrect, are not very informative. Therefore, the standardized coefficients were obtained from the original script using the R Package lm.beta (Behrendt, 2023) and added into a new column in all tables of the appendix (S1 Appendix). Unstandardized coefficients were supplanted by standardized coefficients in the text where applicable. Furthermore, negative and positive values of beta coefficients and t-values in the appendix tables were aligned with those in the text:

page 17: “The overall model was significant F(10, 104) = 2.51 p = .010, R2 = .190 and only fatigue change was found to be a predictor of accuracy change with a negative relationship between accuracy change and fatigue change, β = -.359, t = -2.82, p = .006,”

page 18: “Only age was found to be a predictor of accuracy, β = .372, t = 3.64, p < .001:”

page 18: “only block number found to be a predictor of reaction time, β = -.385, t = -8.94, p < .001,”

page 20: “Pre-task state fatigue was positively related to mental fatigue (β = .187, t = 2.26, p = .026), physical fatigue (β = .351, t = 3.05, p = .003) and general fatigue (β = .731, t = 6.75, p < .001), but was not related to reduced activity (β = .116, t = 1.10, p = .275) or reduced motivation (β = .088, t = .880, p = .380).”

page 20: “Pre-task state energy was negatively related to general fatigue (β = -.672, t = -6.47, p < .001) and reduced activity (β = -.371, t = -3.66, p < .001), whilst showing no relationship to mental fatigue (β = -.090, t = -1.14, p = .256), physical fatigue (β = .211, t = 1.90, p = .060) or reduced motivation (β = .096, t = 1.01, p = .317).”

pages 20-21: “State fatigue change was negatively related to reduced activity (β = -.340, t = -2.31, p = .023), with a small effect size. This meant that participants who had higher reduced activity scores had a smaller increase in their state fatigue during the task. There were no other relationships between state fatigue change and any of the other 4 MFI subscales: mental fatigue (β = -.109, t = -.950, p = .344), physical fatigue (β = -.129, t = -.803, p = .424), general fatigue (β = .098, t = .648, p = .518) or reduced motivation (β = .250, t = 1.80, p = .074). The same model was run to predict state energy change from the five MFI subscales (F(5, 109) = 3.10 p = .012, R2 = .125). Again, reduced activity was also a positive predictor of energy change, β = .377, t = 2.60, p = 0.01 with a small effect size. This showed that participants with higher reduced activity scores reported less energy loss during the task. No relationship was found with the other scales: mental fatigue (β = .046, t = .403, p = .688), physical fatigue (β = .131, t = .829, p = .409), general fatigue (β = -.051, t = -.341, p = .734) and reduced motivation (β = .212, t = 1.55, p = .124).”

Since a new statistical package was utilised to compute the standardised beta coefficients, a reference to it has been added in the relevant part of the results section:

page 12: “‘moments’ [65] and ‘lm.beta’ [66].”

During the process of updating the tables, we noticed minor rounding and transcription inconsistencies when comparing the output of the script to the submitted tables. Therefore, we updated the tables to have the correct readings from the output of the models. It should be noted that these corrections bear no impact on the results of the analysis as reported in the paper.

Discussion:

-I wonder about another potential explanation for the finding that older participants had better nogo performance than younger participants; and that greater fatigue change was associated with poorer nogo performance. Does this mean also that younger people had greater fatigue change with time on task? That is perhaps also somewhat unexpected.

Authors’ response: We thank the reviewer for the valuable reflections about the effects. Answering the first question, it is interesting to note that age failed to be a significant predictor of no-go accuracy change (β = .086, t = .860, p = .391 in S1 Appendix). While we agree with the editor that the findings would point to this added effect, its absence likely arose due to the number of factors included in the various linear models. Since the section in the results already specified which factor affected no-go accuracy change, it would be repetitive to list these again in the discussion.

Another potential factor that should be considered in the discussion is how the at-home, online nature of the task may have resulted in these findings. The use of the remote delivery of the test is discussed, but primarily in the context of its usefulness for engaging in research that uses an online delivery method. However, this approach could also have a different impact on fatigue and performance in younger vs. older individuals, possibly due to novelty of the experience. Perhaps, younger individuals, who may potentially engage in online activities more frequently and proficiently in some regard, may also engage in multitasking more frequently when engaging in analogous activities. Therefore, maintaining attention towards a single, relatively simple task of this type may lead to greater distractibility and be more mentally d

---

## [Editor Report · Decision Letter 2]

25 Apr 2024

Probing sustained attention and fatigue across the lifespan

PONE-D-23-30888R2

Dear Dr. Hanzal,

We’re pleased to inform you that your manuscript has been judged scientifically suitable for publication and will be formally accepted for publication once it meets all outstanding technical requirements.

Kind regards,

Jessica Adrienne Grahn

Academic Editor

PLOS ONE
---

## [Editor Report · Acceptance letter]

30 Apr 2024

PONE-D-23-30888R2 

PLOS ONE

Dear Dr. Hanzal, 

I'm pleased to inform you that your manuscript has been deemed suitable for publication in PLOS ONE. Congratulations! Your manuscript is now being handed over to our production team.

Kind regards, 

on behalf of

Dr Jessica Adrienne Grahn 

Academic Editor

PLOS ONE